# Deep, Skinny Neural Networks are not Universal Approximators

**Jesse Johnson**
Sanofi
jejo.math@gmail.com

## Abstract

In order to choose a neural network architecture that will be effective for a particular modeling problem, one must understand the limitations imposed by each of the potential options. These limitations are typically described in terms of information theoretic bounds, or by comparing the relative complexity needed to approximate example functions between different architectures. In this paper, we examine the topological constraints that the architecture of a neural network imposes on the level sets of all the functions that it is able to approximate. This approach is novel for both the nature of the limitations and the fact that they are independent of network depth for a broad family of activation functions.

## 1 Introduction

Neural networks have become the model of choice in a variety of machine learning applications, due to their flexibility and generality. However, selecting network architectures and other hyperparameters is typically a matter of trial and error. To make the choice of neural network architecture more straightforward, we need to understand the limits of each architecture, both in terms of what kinds of functions any given network architecture can approximate and how those limitations impact its ability to learn functions within those limits.

A number of papers (3; 6; 11; 13) have shown that neural networks with a single hidden layer are a universal approximator, i.e. that they can approximate any continuous function on a compact domain to arbitrary accuracy if the hidden layer is allowed to have an arbitrarily high dimension. In practice, however, the neural networks that have proved most effective tend to have a large number of relatively low-dimensional hidden layers. This raises the question of whether neural networks with an arbitrary number of hidden layers of bounded dimension are also a universal approximator.

In this paper we demonstrate a fairly general limitation on functions that can be approximated with the $L^\infty$ norm on compact subsets of a Euclidean input space by layered, fully-connected feedforward neural networks of arbitrary depth and activation functions from a broad family including sigmoids and ReLus, but with layer widths bounded by the dimension of the input space. By a *layered* network, we mean that hidden nodes are grouped into successive layers and each node is only connected to nodes in the previous layer and the next layer. The constraints on the functions are defined in terms of topological properties of the level sets in the input space.

This analysis is not meant to suggest that deep networks are worse than shallow networks, but rather to better understand how and why they will perform differently on different data sets. In fact, these limitations may be part of the reason deep nets have proven more effective on datasets whose structures are compatible with these limitations.

By a level set, we mean the set of all points in the input space that the model maps to a given value in the output space. For classification models, a level set is just a decision boundary for a particular cutoff. For regression problems, level sets don't have a common interpretation.

The main result of the paper, Theorem 1, states that the deep, skinny neural network architectures described above cannot approximate any function with a level set that is bounded in the input space. This can be rephrased as saying that for every function that can be approximated, every level set must be unbounded, extending off to infinity.

While a number of recent papers have made impressive progress in understanding the limitations of different neural network architectures, this result is notable because it is independent of the number of layers in the network, and because the limitations are defined in terms of a very simple topological property. Topological tools have recently been employed to study the properties of data sets within the field known as Topological Data Analysis (9), but this paper exploits topological ideas to examine the topology of the models themselves. By demonstrating topological constraints on a widely used family of models, we suggest that there is further potential to apply topological ideas to understand the strengths and weaknesses of algorithms and methodologies across machine learning.

After discussing the context and related work in Section 2, we introduce the basic definitions and notation in Section 3, then state the main Theorem and outline the proof in Section 4. The detailed proof is presented in Sections 5 and 6. We present experimental results that demonstrate the constraints in Section 7, then in Section 8 we present conclusions from this work.

## 2 RELATED WORK

A number of papers have demonstrated limitations on the functions that can be approximated by neural networks with particular architectures (2; 12; 14; 15; 18; 19; 21; 22; 23; 24; 25; 27; 29; 32). These are typically presented as asymptotic bounds on the size of network needed to approximate any function in a given family to a given $\varepsilon$.

Lu et al (17) gave the first non-approximation result that is independent of complexity, showing that there are functions that no ReLu-based deep network of width equal to the dimension of the input space can approximate, no matter how deep. However, they consider convergence in terms of the $L^1$ norm on the entire space $\mathbb{R}^n$ rather than $L^\infty$ on a compact subset. This is a much stricter definition than the one used in this paper so even for ReLu networks, Theorem 1 is a stronger result.

The closest existing result to Theorem 1 is a recent paper by Nguyen, Mukkamala and Hein (26) which shows that for multi-label classification problems defined by an argmax condition on a higher-dimensional output function, if all the hidden layers of a neural network have dimension less than or equal to the input dimension then the region defining each class must be connected. The result applies to one-to-one activation functions, but could probably be extended to the family of activation functions in this paper by a similar limiting argument.

Universality results have been proved for a number of variants of the networks described in Theorem 1. Rojas (28) showed that any two discrete classes of points can be separated by a decision boundary of a function defined by a deep, skinny network in which each layer has a single perceptron that is connected both to the previous layer and to the input layer. Because of the connections back to the input space, such a network is not layered as defined above, so Theorem 1 doesn't contradict this result. In fact, to carry out Rojas' construction with a layered feed-forward network, you would need to put all the perceptrons in a single hidden layer.

Sutskever and Hinton (30) showed that deep belief networks whose hidden layers have the same dimension as the input space can approximate any function over binary vectors. This binary input space can be interpreted as a discrete subset of Euclidean space. So while Theorem 1 does not apply to belief networks, it's worth noting that any function on a discrete set can be extended to the full space in such a way that the resulting function satisfies the constraints in Theorem 1.

This unexpected constraint on skinny deep nets raises the question of whether such networks are so practically effective despite being more restrictive than wide networks, or because of it. Lin, Tegmark and Rolnick (16) showed that for data sets with information-theoretic properties that are common in physics and elsewhere, deep networks are more efficient than shallow networks. This may be because such networks are restricted to a smaller search space concentrated around functions that model shapes of data that are more likely to appear in practice. Such a conclusion would be consistent with a number of papers showing that there are functions defined by deep networks that can only by approximated by shallow networks with asymptotically much larger number of nodes (4; 7; 10; 20; 31).

A slightly different phenomenon has been observed for recurrent neural networks, which are universal approximators of dynamic systems (8). In this setting, Collins, Sohl-Dickstein and Sussillo (5) showed that many differences that have been reported on the performance of RNNs are due to their

training effectiveness, rather than the expressiveness of the networks. In other words, the effectiveness of a given family of models appears to have less to do with whether it includes an accurate model, and more to do with whether a model search algorithm like gradient descent is likely to find an accurate model within the search space of possible models.

## 3 TERMINOLOGY AND NOTATION

A *model family* is a subset $M$ of the space $\mathcal{C}(\mathbb{R}^n, \mathbb{R}^m)$ of continuous functions from input space $\mathbb{R}^n$ to output space $\mathbb{R}^m$. For parametric models, this subset is typically defined as the image of a map $\mathbb{R}^k \to \mathcal{C}(\mathbb{R}^n, \mathbb{R}^m)$, where $\mathbb{R}^k$ is the parameter space. A non-parametric model family is typically the union of a countably infinite collection of parametric model families. We will not distinguish between parametric and non-parametric families in this section.

Given a function $g : \mathbb{R}^n \to \mathbb{R}^m$, a compact subset $A \subset \mathbb{R}^n$ and a value $\epsilon > 0$, we will say that a second function $f$ $(\epsilon, A)$-*approximates* $g$ if for every $x \in A$, we have $|f(x) - g(x)| < \epsilon$. Similarly, we will say that a model family $M$ $(\epsilon, A)$-*approximates* $g$ if there is a function $f$ in $M$ that $(\epsilon, A)$-approximates $g$.

More generally, we will say that $M$ *approximates* $f$ if for every compact $A \subset \mathbb{R}^n$ and value $\epsilon > 0$ there is a function $f$ in $M$ that $(\epsilon, A)$-approximates $g$. This is equivalent to the statement that there is a sequence of functions $f_i \in M$ that converges pointwise (though not necessarily uniformly) to $g$ on all of $\mathbb{R}^n$. However, we will use the $(\epsilon, A)$ definition throughout this paper.

We'll describe families of layered neural networks with the following notation: Given an activation function $\varphi : \mathbb{R} \to \mathbb{R}$ and a finite sequence of positive integers $n_0, n_1, \ldots, n_\kappa$, let $\mathcal{N}_{\varphi, n_0, n_1, \ldots, n_\kappa}$ be the family of functions defined by a layered feed-forward neural network with $n_0$ inputs, $n_\kappa$ outputs and fully connected hidden layers of width $n_1, \ldots, n_{\kappa-1}$.

With this terminology, Hornik et al's results can be restated as saying that the (non-parametric) model family defined as the union of all families $\mathcal{N}_{\varphi, n_0, n_1, 1}$ approximates any continuous function. (Here, $\kappa = 2$ and $n_2 = 1$.)

We're interested in deep networks with bounded dimensional layers, so we'll let $\mathcal{N}_{\varphi, n}^*$ be the union of all the model families $\mathcal{N}_{\varphi, n_0, n_1, \ldots, n_{\kappa-1}, 1}$ such that $n_i \leq n$ for all $i < \kappa$.

For the main result, we will restrict our attention to a fairly large family of activation functions. We will say that an activation function $\varphi$ is *uniformly approximated by one-to-one functions* if there is a sequence of continuous, one-to-one functions that converge to $\varphi$ uniformly (not just pointwise).

Note that if the activation function is itself one-to-one (such as a sigmoid) then we can let every function in the sequence be $\varphi$ and it will converge uniformly. For the ReLu function, we need to replace the the large horizontal portion with a function such as $\frac{1}{n} \arctan(x)$. Since this function is one-to-one and negative for $x < 0$, each function in this sequence will be one-to-one. Since it's bounded between $-\frac{1}{n}$ and $0$, the sequence will converge uniformly to the ReLu function.

## 4 OUTLINE OF THE MAIN RESULT

The main result of the paper is a topological constraint on the level sets of any function in the family of models $\mathcal{N}_{\varphi, n}^*$. To understand this constraint, recall that in topology, a set $C$ is *path connected* if any two points in $C$ are connected by a continuous path within $C$. A *path component* of a set $A$ is a subset $C \subset A$ that is connected, but is not a proper subset of a larger connected subset of $A$.

**Definition 1.** We will say that a function $f : \mathbb{R}^n \to \mathbb{R}$ has *unbounded level components* if for every $y \in \mathbb{R}$, every path component of $f^{-1}(y)$ is unbounded.

The main result of this paper states that deep, skinny neural networks can only approximate functions with unbounded level components. Note that this definition is stricter than just requiring that every level set be bounded. The stricter definition in terms of path components guarantees that the property is preserved by limits, a fact that we will prove, then use in the proof of Theorem 1. Just having bounded level sets is not preserved under limits.

**Theorem 1.** *For any integer $n \geq 2$ and uniformly continuous activation function $\varphi : \mathbb{R} \to \mathbb{R}$ that can be approximated by one-to-one functions, the family of layered feed-forward neural networks*

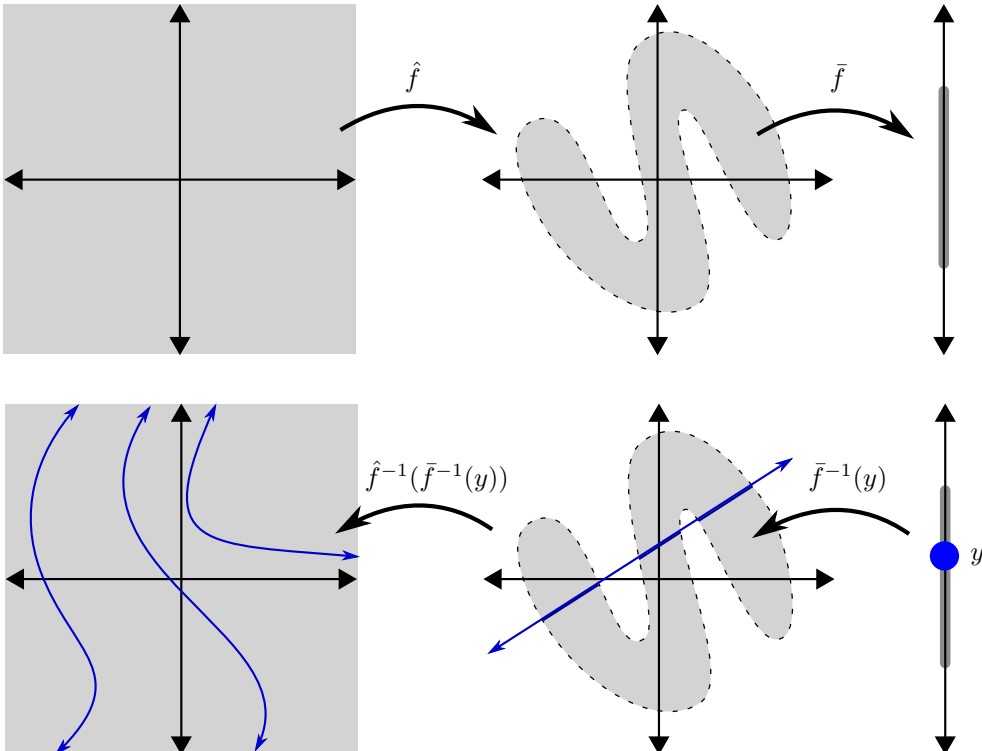

Figure 1: A generic function defined by a deep network is the composition of a one-to-one function and a linear function. Each level set is therefore homeomorphic to the intersection of an open topological ball with an $n-1$-dimensional hyperplane, which forces it to be unbounded.

*with input dimension $n$ in which each hidden layer has dimension at most $n$ cannot approximate any function with a level set containing a bounded path component.*

The proof of Theorem 1 consists of two steps. In the first step, described in Section 5, we examine the family of functions defined by deep, skinny neural networks in which the activation is one-to-one and the transition matrices are all non-singular.

We prove two results about this smaller family of functions: First, Lemma 2 states that any function that can be approximated by $\mathcal{N}^*_{\varphi,n}$ can be approximated by functions in this smaller family. This is fairly immediate from the assumptions on $\varphi$ and the fact that singular transition matrices can be approximated by non-singular ones.

Second, Lemma 4 states that the level sets of these functions have unbounded level components. The proof of this Lemma is, in many ways, the core argument of the paper and is illustrated in Figure 1. The idea is that any function in this smaller family can be written as a composition of a one-to-one function and a linear projection, as in the top row of the Figure.

As suggested in the bottom row, this implies that each level set/decision boundary in the full function is defined by the intersection of the image of the one-to-one function (the gray patch in the middle) with a hyperplane that maps to a single point in the second function. Intuitively, this intersection extends out to the edges of the gray blob, so its preimage in the original space must extend out to infinity in Euclidean space, i.e. it must be unbounded.

The second part of the proof of Theorem 1, described in Section 5, is Lemma 5 which states that the limit of functions with unbounded level components also has unbounded level components. This is

a subtle technical argument, though it should be intuitively unsurprising that unbounded sets cannot converge to bounded sets.

The proof of Theorem 1 is the concatenation of these three Lemmas: If a function can be approximated by $\mathcal{N}^*_{\varphi,n}$ then it can be approximated by the smaller model family (Lemma 2), so it can be approximated by functions with unbounded level components (Lemma 4), so it must also have unbounded level components (Lemma 5).

## 5    CHARACTERIZING LEVEL SETS OF "GENERIC" NEURAL NET FUNCTIONS

We will say that a function in $\mathcal{N}^*_{\varphi,n}$ is *non-singular* if $\varphi$ is continuous and one-to-one, $n_i = n$ for all $i < k$ and the matrix defined by the weights between each pair of layers is nonsingular. Note that if $\varphi$ is not one-to-one, then $\mathcal{N}^*_{\varphi,n}$ will not contain any non-singular functions. If it is one-to-one then $\mathcal{N}^*_{\varphi,n}$ will contain a mix of singular and non-singular functions.

Define the *model family of non-singular functions* $\hat{\mathcal{N}}_n$ to be the union of all non-singular functions in families $\mathcal{N}^*_{\varphi,n}$ for all activation functions $\varphi$ and a fixed $n$.

**Lemma 2.** *If $g$ is approximated by $\mathcal{N}^*_{\varphi,n}$ for some continuous activation function $\varphi$ that can be uniformly approximated by one-to-one functions then it is approximated by $\hat{\mathcal{N}}_n$.*

To prove this Lemma, we will employ a technical result from point-set topology, relying on the fact that a function in $\mathcal{N}^*_{\varphi,n}$ can be written as a composition of linear functions defined by the weights between successive layers, and non-linear functions defined by the activation function $\varphi$.

**Lemma 3.** *Assume $f : \mathbb{R}^n \to \mathbb{R}^m$ is a function that can be written as a composition $f = f_\kappa \circ \cdots \circ f_1 \circ f_0$ where each $f_i : \mathbb{R}^{n_i} \to \mathbb{R}^{n_{i+1}}$ is a continuous function. Let $A \subset \mathbb{R}^n$ be a compact subset and choose $\varepsilon > 0$.*

*Then there is a compact subset $A_i \subset \mathbb{R}^{n_i}$ for each $i$ and $\delta > 0$ such that if $g_i : \mathbb{R}^{n_i} \to \mathbb{R}^{n_{i+1}}$ is a function that $(\delta, A_i)$-approximates $f_i$ for each $i$ then the composition $g = g_\kappa \circ \cdots \circ g_1 \circ g_0$ will $(\varepsilon, A)$-approximate $g$.*

One can prove Lemma 3 by induction on the number of functions in the composition, choosing each $A_i \subset \mathbb{R}^{n_i}$ to be a closed $\varepsilon$-neighborhood of the image of $A$ in the composition up to $i$. For each new function, the $\delta$ on the compact set tells you what $\delta$ you need to choose for the composition of the preceding functions. We will not include the details here.

*Proof of Lemma 2.* We'll prove this Lemma by showing that $\hat{\mathcal{N}}_n$ approximates any given function in $\mathcal{N}^*_{\varphi,n}$. Then, given $\varepsilon > 0$, a compact set $A \subset \mathbb{R}^n$ and a function $g$ that is approximated by $\mathcal{N}^*_{\varphi,n}$, we can choose a function $f \in \mathcal{N}^*_{\varphi,n}$ that $(\varepsilon/2, A)$-approximates $g$ and a function in $\hat{\mathcal{N}}_n$ that $(\varepsilon/2, A)$-approximates $f$.

So we will reset the notation, let $g$ be a function in $\mathcal{N}^*_{\varphi,n}$, let $A \subset \mathbb{R}^n$ be a compact subset and choose $\varepsilon > 0$. As noted above, $g$ is a composition $g = \nu_\kappa \circ \ell_\kappa \circ \cdots \circ \nu_0 \circ \ell_0$ where each $\ell_i$ is a linear function defined by the weights between consecutive layers and each $\nu_i$ is a nonlinear function defined by a direct product of the activation function $\varphi$.

If any of the hidden layers in the network defining $g$ have dimension strictly less than $n$ then we can define the same function with a network in which that layer has dimension exactly $n$, but the weights in and out of the added neurons are all zero. Therefore, we can assume without loss of generality that all the hidden layers in $g$ have dimension exactly $n$, though the linear functions may be singular.

Let $\{A_i\}$ and $\delta > 0$ be as defined by Lemma 3. We want to find functions $\hat{\nu}_i$ and $\hat{\ell}_i$ that $(\delta, A_i)$-approximate each $\nu_i$ and $\ell_i$ and whose composition is in $\hat{\mathcal{N}}_n$.

For the composition to be in $\hat{\mathcal{N}}_n$, we need each $\hat{\ell}_i$ to be non-singular. If $\ell_i$ is already non-singular, then we choose $\hat{\ell}_i = \ell_i$. Otherwise, we can perturb the weights that define the linear map $\ell_i$ by an arbitrarily small amount to make it non-singular. In particular, we can choose this arbitrarily small amount to be small enough that the function values change by less than $\delta$ on $A_i$.

Similarly, we want each $\hat{\nu}_i$ to be a direct product of a continuous, one-to-one activation functions. By assumption, $\varphi$ can be approximated by such functions and we can choose the tolerance for this approximation to be small enough that $\hat{\nu}_i$ $(\delta, A_i)$-approximates $\nu_i$. In fact, we can choose a single activation function for all the nonlinear layers, on each corresponding compact set.

Thus we can choose each $\hat{\ell}_i$ and an activation function $\hat{\varphi}$ that defines all the functions $\nu_i$, so that the composition is in $\hat{\mathcal{N}}_n$ and, by Lemma 3, the composition $(\varepsilon, A)$-approximates $g$. □

# 6 CHARACTERIZING LEVEL SETS IN A LIMIT OF FUNCTIONS.

Lemma 2 implies that if $\mathcal{N}_{\varphi,n}^*$ is universal then so is $\hat{\mathcal{N}}_n$. So to prove Theorem 1, we will show that every function in $\hat{\mathcal{N}}_n$ has level sets with only unbounded components, then show that this property extends to any function that it approximates.

**Lemma 4.** *If $f$ is a function in $\hat{\mathcal{N}}_n$ then every level set $f^{-1}(y)$ is homeomorphic to an open (possibly empty) subset of $\mathbb{R}^{n-1}$. This implies that $f$ has unbounded level components.*

*Proof.* Assume $f$ is a non-singular function in $\hat{\mathcal{N}}_n$, where $\varphi$ is continuous and one-to-one. Let $\hat{f} : \mathbb{R}^n \to \mathbb{R}^n$ be the function defined by all but the last layer of the network. Let $\bar{f} : \mathbb{R}^n \to \mathbb{R}$ be the function defined by the map from the last hidden layer to the final output layer so that $f = \bar{f} \circ \hat{f}$.

The function $\hat{f}$ is a composition of the linear functions defined by the network weights and the non-linear function at each step defined by applying the activation function to each dimension. Because $f$ is nonsingular, the linear functions are all one-to one. Because $\varphi$ is continuous and one-to-one, so are all the non-linear functions. Thus the composition $\hat{f}$ is also one-to-one, and therefore a homeomorphism from $\mathbb{R}^n$ onto its image $I_{\hat{f}}$. Since $\mathbb{R}^n$ is homeomorphic to an open $n$-dimensional ball, $I_{\hat{f}}$ is an open subset of $\mathbb{R}^n$, as indicated in the top row of Figure 1.

The function $\bar{f}$ is the composition of a linear function to $\mathbb{R}$ with $\varphi$, which is one-to-one by assumption. So the preimage $\bar{f}^{-1}(y)$ for any $y \in \mathbb{R}$ is an $(n-1)$-dimensional plane in $\mathbb{R}^n$. The preimage $f^{-1}(y)$ is the preimage in $\hat{f}$ of this $(n-1)$-dimensional plane, or rather the preimage of the intersection $I_{\hat{f}} \cap \bar{f}^{-1}(y)$, as indicated in the bottom right/center of the Figure. Since $I_{\hat{f}}$ is open as a subset of $\mathbb{R}^n$, the intersection is open as a subset of $\bar{f}^{-1}(y)$.

Since $\hat{f}$ is one-to-one, its restriction to this preimage (shown on the bottom left of the Figure) is a homeomorphism from $f^{-1}(y)$ to this open subset of the $(n-1)$-dimensional plane $\bar{f}^{-1}(y)$. Thus $f^{-1}(y)$ is homeomorphic to an open subset of $\mathbb{R}^{n-1}$.

Finally, recall that the preimage in a continuous function of a closed set is closed, so $f^{-1}(y)$ is closed as a subset of $\mathbb{R}^n$. If it were also bounded, then it would be compact. However, the only compact, open subset of $\mathbb{R}^{n-1}$ is the empty set, so $f^{-1}(y)$ is either unbounded or empty. Since each path component of a subset of $\mathbb{R}^{n-1}$ is by definition non-empty, this proves that any component of $f$ is unbounded. □

All that remains is to show that this property extends to the functions that $\hat{\mathcal{N}}_n$ approximates.

**Lemma 5.** *If $M$ is a model family in which every function has unbounded level components then any function approximated by $M$ has unbounded level components.*

*Proof.* Let $g : \mathbb{R}^n \to \mathbb{R}$ be a function with a level set $g^{-1}(y)$ containing a bounded path component $C$. Note that level sets are closed as subsets of $\mathbb{R}^n$ and bounded, closed sets are compact so $C$ is compact. We can therefore choose a value $\mu$ such that any point of $g^{-1}(y)$ outside of $C$ is distance greater than $\mu$ from every point in $C$.

Let $\eta_C$ be the set of all points that are distance strictly less than $\mu/2$ from $C$. This is an open subset of $\mathbb{R}^n$, shown as the shaded region in the center of Figure 2, and we will let $F$ be the frontier of $\eta_C$ – the set of all points that are limit points of both $\eta_C$ and limit points of its complement.

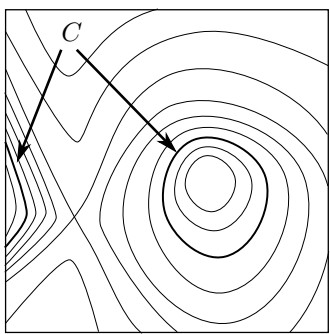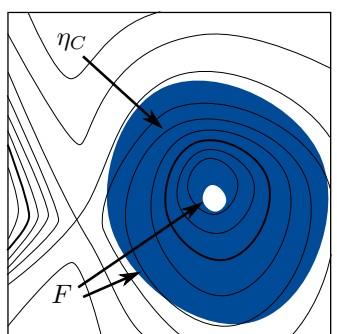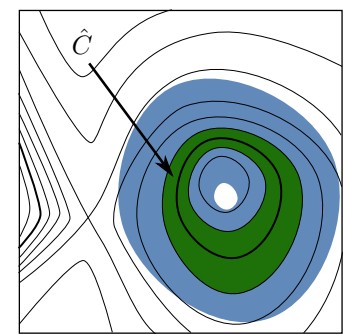

Figure 2: In the proof of Lemma 2, we choose $\varepsilon$ by finding an intercal $U = (y - \varepsilon, y + \varepsilon)$ whose preimage contains a bounded component $\hat{C}$.

By construction, every point in $F$ is distance $\mu/2$ from $C$ so $F$ is disjoint from $C$. Moreover, since every point in $g^{-1}(y) \setminus C$ is distance at least $\mu$ from $C$, $F$ is disjoint from the rest of $g^{-1}(y)$ as well, so $y$ is in the complement of $g(F)$.

The frontier is the intersection of two closed sets, so $F$ is closed. It's also bounded, since all points are a bounded distance from $C$, so $F$ is compact. This implies that $g(F)$ is a compact subset of $\mathbb{R}$, so its complement is open. Since $y$ is in the complement of $g(F)$, this means that there is an open interval $U = (y - \varepsilon, y + \varepsilon)$ that is disjoint from $g(F)$.

Let $\hat{C}$ be the component of $g^{-1}(U)$ that contains $C$, as indicated on the right of the Figure. Note that this set intersects $\eta_C$ but is disjoint from its frontier. So $\hat{C}$ must be contained in $\eta_C$, and is therefore bounded as well. In particular, each level set that intersects $\hat{C}$ has a compact component in $\hat{C}$.

Let $x$ be a point in $C \subset \hat{C}$. Since $\hat{C}$ is bounded, there is a value $r$ such that every point in $\hat{C}$ is distance at most $r$ from $x$.

Assume for contradiction that $g$ is approximated by a model family $M$ in which each function has unbounded level components. Choose $R > r$ and let $B_R(x)$ be a closed ball of radius $R$, centered at $x$. Because this is a compact set and $g$ is approximated by $M$, we can choose a function $f \in M$ that $(\varepsilon/2, B_R(x))$-approximates $g$.

Then $|f(x) - g(x)| < \varepsilon/2$ so $f(x) \in [y - \varepsilon/2, y + \varepsilon/2] \subset U$ and we will define $y' = f(x)$.

Since $f \in M$, every path component of $f^{-1}(y')$ is unbounded, so there is a path $\ell \subset f^{-1}(y')$ from $x$ to a point that is distance $R$ from $x$. If $\ell$ passes outside of $B_R(x))$, we can replace $\ell$ with the component of $\ell \cap B_R(x))$ containing $x$ to ensure that $\ell$ stays inside of $B_R(x))$, but still reaches a point that is distance $R$ from $x$.

Since every point $x'' \in \ell$ is contained in $B_R(x)$, we have $|f(x'') - g(x'')| < \varepsilon/2$. This implies $g(x'') \in [y - \varepsilon, y + \varepsilon] = U$ so the path is contained in $g^{-1}(U)$, and thus in the path component $\hat{C}$ of $g^{-1}(U)$.

However, by construction the path $\ell$ ends at a point whose distance from $x$ is $R > r$, contradicting the assumption that every point in $\hat{C}$ is distance at most $r$ from $x$. This contradiction proves that $g$ cannot be approximated by a model family $M$ in which each function has unbounded level components. $\qquad\square$

*Proof of Theorem 1.* Let $g$ be a function that is approximated by $\mathcal{N}^*_{\varphi,n}$, where $\varphi$ is a continuous activation function that can be uniformly approximated by one-to-one functions.

By Lemma 2, since $g$ is approximated by $\mathcal{N}^*_{\varphi,n}$, it must also be approximated by $\hat{\mathcal{N}}_n$. By Lemma 4, every function in $\hat{\mathcal{N}}_n$ has bounded level components, so Lemma 5 implies that every function that this family approximates has unbounded level sets. Therefore $g$ has unbounded level sets. $\qquad\square$

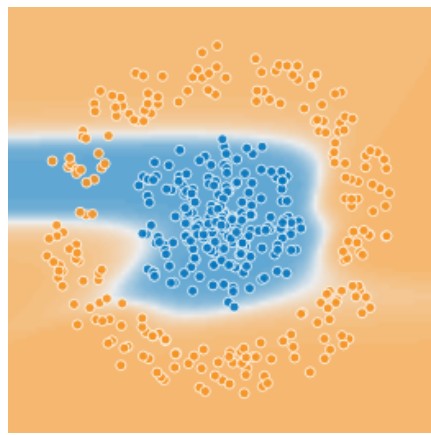 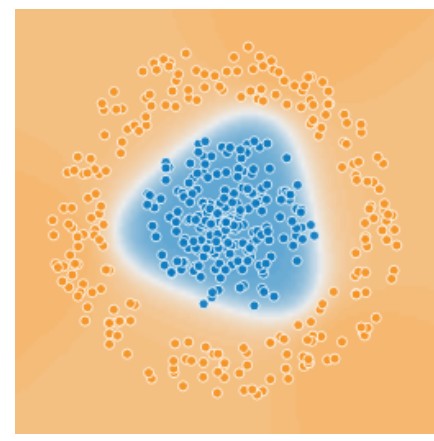

(a) The decision boundary learned with six, two-dimensional hidden layers is an unbounded curve that extends outside the region visible in the image.

(b) A network with a single three-dimensional hidden layer learns a bounded decision boundary relatively easily.

## 7 EXPERIMENTS

To demonstrate the effect of Theorem 1, we used the TensorFlow Neural Network Playground (1) to train two different networks on a standard synthetic dataset with one class centered at the origin of the two-dimensional plane, and the other class forming a ring around it. We trained two neural networks and examined the plot of the resulting functions to characterize the level sets/decision boundaries. In these plots, the decision boundary is visible as the white region between the blue and orange regions defining the two labels.

The first network has six two-dimensional hidden layers, the maximum number of layers allowed in the webapp. As shown in Figure 3a, the decision boundary is an unbounded curve that extends beyond the region containing all the data points. The ideal decision boundary between the two classes of points would be a (bounded) loop around the blue points in the middle, but Theorem 1 proves that such a network cannot approximate a function with such a level set. A decision boundary such as the one shown in the Figure is as close as it can get. The extra hidden layers allow the decision boundary to curve around and minimize the neck of the blue region, but they do not allow it to pinch off completely.

The second network has a single hidden layer of dimension three - one more than that of the input space. As shown in Figure 3b, the decision boundary for the learned function is a loop that approximates the ideal decision boundary closely. It comes from the three lines defined by the hidden nodes, which make a triangle that gets rounded off by the activation function. Increasing the dimension of the hidden layer would make the decision boundary rounder, though in this case the model doesn't need the extra flexibility.

Note that this example generalizes to any dimension $n$, though without the ability to directly graph the results. In other words, for any Euclidean input space of dimension $n$, a sigmoid neural network with one hidden layer of dimension $n + 1$ can define a function that cannot be approximated by any deep network with an arbitrary number of hidden layers of dimension at most $n$. In fact, this will be the case for any activation function that is bounded above or below, though we will not include the details of the argument here.

## 8 CONCLUSION

In this paper, we describe topological limitations on the types of functions that can be approximated by deep, skinny neural networks, independent of the number of hidden layers. We prove the result using standard set theoretic topology, then present examples that visually demonstrate the result.

This complements a body of existing literature that has demonstrated various limitations on neural networks that typically take a very different form and are expressed in terms of asymptotic network complexity. We expect that there is a great deal of remaining potential to explore further topological constraints on families of models, and to determine to what extent these topological constraints are simply a different way of describing more fundamental ideas that have been independently demonstrated elsewhere in other frameworks such as information theory.

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
