# OpenReview forum: "Deep, Skinny Neural Networks are not Universal Approximators"
_ICLR.cc/2019/Conference_

### Official Review · AnonReviewer2 · 2018-10-31
**An interesting proof on approximation capabilities of deep skinny neural networks**

**Rating:** 7
**Confidence:** 4

**Review:**

This paper shows that deep "narrow" neural networks (i.e. all hidden layers have maximum width at most the input dimension) with a variety of activation functions, including ReLU and sigmoid, can only learn functions with unbounded level set components, and thus cannot be a universal approximator. This complements previous work, such as Nguyen et. al 2018 which study connectivity of decision regions and Lu et. al 2017 on ReLU networks in different ways.

Overall the paper is clearly written and technically sound. The result itself may not be super novel as noted in the related work but it's still a strict improvement over previous results which is often constrained to ReLU activation function. Moreover, the proofs of this paper are really nice and elegant. Compared to other work on approximation capability of neural networks, it can tell us in a more intuitive way and explicitly which class of functions/problems cannot be learned by neural networks if none of their layers have more neurons than the input dimension, which might be helpful in practice. Given the fact that there are not many previous work that take a similar approach in this direction, I'm happy to vote for accepting this paper.

Minor comments:
The proof of Lemma 3 should be given for completeness. I guess this can be done more easily by setting delta=epsilon, A_0=A and A_{i+1}=epsilon-neighborhood of f_i(A_i)?
page7: the square brackets in "...g(x'')=[y-epsilon,y+epsilon]..." should be open brackets.
page7:"By Lemma 4, every function in N_n has bounded level components..." -> "..unbounded..."

---

### Official Review · AnonReviewer1 · 2018-11-02
**A solid contribution to a relatively underexplored area of machine learning**

**Rating:** 8
**Confidence:** 4

**Review:**

This is a very nice paper contributing to what I consider a relatively underexplored but potentially very promising research direction. The title of the paper in my opinion undersells the result which is not only that "deep skinny neural networks" are not universal approximators, but that the class of functions which cannot be approximated includes a set of practically relevant classifiers as illustrated by the figure on page 8. The presentation is extremely clear with helpful illustrations and toy but insightful experiments.

My current rating of this paper is based on assuming that the following concerns will be addressed. I will adjust the score accordingly after authors' reply.



Main:

- A very similar result can be found in Theorem 7 of Beise et al.'s "On decision regions of narrow deep neural networks" from July 2018 ( https://arxiv.org/abs/1807.01194 )
	Some differences:

		- The other paper considers connected whereas this paper considers path-connected components (the former is more general).
		- The other paper only considers multi-label classification, this paper is relevant to all classification and regression problems (the latter is more general).
		- The other paper requires that the activation function is "strictly monotonic or ReLU" whereas this paper allows "uniformly approximable with one-to-one functions" activations (the latter is more general).

	The result in this paper seems slightly more general but largely similar. Can you please comment on the differences/relation to the other paper?


- Proof of Lemma 4:  "Thus the composition \hat{f} is also one-to-one, and therefore a homeomorphism from R^n onto its image I_{\hat{f}}". Is it not necessary that \hat{f} has a continuous inverse in order to be a homeomorphism? I do not immediately see whether the class of activation functions considered in this paper implies that this condition is satisfied. Please clarify.



Minor:

- Proof of Lemma 5: It seems g is assumed to be continuous at several places (e.g. "... level sets of are closed as subsets of R^n ..." seems to assume that pre-image of a closed set under g is closed, or later "This implies g(F) is a compact subset of R ..."). Perhaps you are assuming that M is a set of continuous functions and using the fact that uniform limit of continuous functions is continuous? Please clarify.

- On p.4: "This is fairly immediate from the assumptions on \varphi and the fact that singular transition matrices can be approximated by non-singular ones." Is the second part of the sentence using the assumption that the input space is compact? Please clarify.

- Second line in Section 5: i < k should probably be i < \kappa.

---

### Official Review · AnonReviewer3 · 2018-11-05
**Review of "Deep, Skinny Neural Networks are not Universal Approximators"**

**Rating:** 6
**Confidence:** 4

**Review:**

This paper proves a theoretical limitation of narrow-and-deep neural networks. It shows that, for any function that can be approximated by such networks, its level set (or decision boundary for binary classification) must be unbounded. The conclusion means that if some problem's decision boundary is a closed set, then it cannot be represented by such narrow networks.

The intuition is relatively simple. Under the assumptions of the paper, the neural network can always be approximated by a one-to-one mapping followed by a linear projection. The image of the one-to-one mapping is homeomorphic to R^n, so that it must be an open topological ball. The intersection of this open ball with a linear hyperplane must include the boundary of the ball, thus it extends to infinity in the original input space. The critical assumptions here, which guarantees the one-to-one property of the network, are: 1) the network is narrow, and 2) the activation function can be approximated by a one-to-one function.

The authors claim that 2) captures a large family of activation functions. However, it does exclude some popular activation families, such as the polynomial activation, which were proven effective in multiple areas. As a concrete example, the simple function f(x1,x2) = x_1^2 + x_2^2 has bounded level sets, but it can be represented by a narrow 2-layer neural network with the quadratic activation.

Overall, I feel that the result is interesting but it depends on a strong assumption and doesn't capture all interesting cases. It is also not clear how this theoretical result can shed insight on the empirical study of neural networks.

---

> ### Author Response · Authors · 2018-12-07
> **On the narrowness of the result**
>
> It's true that there are many activation functions that the result doesn't apply to, and in fact isn't true for. The selling point isn't the generality of the result, but rather the novelty of the approach and the potential it suggests for future work that would be more general.

---

### Public Comment · (anonymous) · 2018-10-05
**Amazing paper**

I love this paper. Please keep up the good work.

---

### Public Comment · ~Guido_Novati1 · 2018-10-13
**Two comments**

1) Can the Authors comment on how the contribution of this work differs from [1]?

2) These conclusions do not seem to hold for skinny deep NN with residual connections [2]. How could Theorem 1 be modified to include this evidence?

[1] http://papers.nips.cc/paper/5422-on-the-number-of-linear-regions-of-deep-neural-networks

[2] https://github.com/novatig/playground

---

> ### Author Response · Authors · 2018-12-07
> **Response to "Two comments"**
>
> The major difference between this paper and [1] is that while [1] describes the rate at which the complexity of model functions can increase as the network architecture increases, the present paper describes a limitation that is independent of network complexity within a family of neural networks.
>
> These results most likely don't hold for networks with residual connections, since from the topological perspective the residual connections effectively increase the dimension of the maps between layers.

---

### Comment · Area_Chair1 · 2018-12-16
**narrow DBNs**

The referenced paper on narrow belief networks uses layers of width n+1 and poses size n as as an open problem. A later work by Le Roux and Bengio obtained width n.

---

### Meta-Review · Area_Chair1 · 2018-12-16
**Analysis of obstructions in skinny networks**

**Confidence:** 4
**Recommendation:** Accept (Poster)

**Metareview:**

The paper shows limitations on the types of functions that can be represented by deep skinny networks for certain classes of activation functions, independently of the number of layers. With many other works discussing capabilities but not limitations, the paper contributes to a relatively underexplored topic.

The settings capture a large family of activation functions, but exclude others, such as polynomial activations, for which the considered type of obstructions would not apply. Also a concern is raised about it not being clear how this theoretical result can shed insight on the empirical study of neural networks.

The authors have responded to some of the comments of the reviewers, but not to all comments, in particular comments of reviewer 1, who's positive review is conditional on the authors addressing some points.

The reviewers are all confident and are moderately positive, positive, or very positive about this paper.